# Sentiment Analysis in Understanding the Potential of Online News in the Public Health Crisis Response

**DOI:** 10.3390/ijerph192416801

**Published:** 2022-12-14

**Authors:** Thiago Marques, Sidemar Cezário, Juciano Lacerda, Rafael Pinto, Lyrene Silva, Orivaldo Santana, Anna Giselle Ribeiro, Agnaldo Souza Cruz, Angélica Espinosa Miranda, Aedê Cadaxa, Lucía Sanjuán Núñez, Hugo Gonçalo Oliveira, Rifat Atun, Ricardo Valentim

**Affiliations:** 1Department of Informatics and Applied Mathematics, Federal University of Rio Grande do Norte, Natal 59078-970, Brazil; 2Department of Social Communication, Federal University of Rio Grande do Norte, Natal 59072-970, Brazil; 3Laboratory of Technological Innovation in Health (LAIS), Federal University of Rio Grande do Norte, Natal 59010-090, Brazil; 4Information Systems Coordination, Federal Institute of Rio Grande do Norte, Natal 59015-300, Brazil; 5School of Science and Technology, Federal University of Rio Grande do Norte, Natal 59078-970, Brazil; 6Ministry of Health, Brasília 70070-600, Brazil; 7Postgraduate Program in Infectious Diseases, Federal University of Espírito Santo, Vitória 29075-910, Brazil; 8Department of Social and Cultural Anthropology, Autonomous University of Barcelona, 08193 Barcelona, Spain; 9Centre for Informatics and Systems of the University of Coimbra (CISUC), Department of Informatics Engineering (DEI), University of Coimbra, 3030-290 Coimbra, Portugal; 10Health Systems Innovation Laboratory, Harvard TH Chan School Public Health, Harvard University, Boston, MA 02115, USA; 11Department of Global Health and Population, Harvard TH Chan School Public Health, Harvard University, Boston, MA 02115, USA; 12Department of Biomedical Engineering, Federal University of Rio Grande do Norte, Natal 59628-330, Brazil

**Keywords:** sentiment analysis, public health, digital solution, online news, public policy

## Abstract

This study analyzes online news disseminated throughout the pre-, during-, and post-intervention periods of the “Syphilis No!” Project, which was developed in Brazil between November 2018 and March 2019. We investigated the influence of sentiment aspects of news to explore their possible relationships with syphilis testing data in response to the syphilis epidemic in Brazil. A dictionary-based technique (VADER) was chosen to perform sentiment analysis considering the Brazilian Portuguese language. Finally, the data collected were used in statistical tests to obtain other indicators, such as correlation and distribution analysis. Of the 627 news items, 198 (31.58%) were classified as a sentiment of security (TP2; stands for the news type 2), whereas 429 (68.42%) were classified as sentiments that instilled vulnerability (TP3; stands for the news type 3). The correlation between the number of syphilis tests and the number of news types TP2 and TP3 was verified from (i) 2015 to 2017 and (ii) 2018 to 2019. For the TP2 type news, in all periods, the *p*-values were greater than 0.05, thus generating inconclusive results. From 2015 to 2017, there was an ρ = 0.33 correlation between TP3 news and testing data (*p*-value = 0.04); the years 2018 and 2019 presented a ρ = 0.67 correlation between TP3 news and the number of syphilis tests performed per month, with *p*-value = 0.0003. In addition, Granger’s test was performed between TP3 news and syphilis testing, which resulted in a *p*-value = 0.002, thus indicating the existence of Granger causality between these time series. By applying natural language processing to sentiment and informational content analysis of public health campaigns, it was found that the most substantial increase in testing was strongly related to attitude-inducing content (TP3).

## 1. Introduction

According to the Pan American Health Organization (PAHO) [1], syphilis is a Sexually Transmitted Infection (STI) caused by the bacterium *Treponema pallidum*. Left untreated or inadequately treated, the disease can lead to severe health problems and even cause death in infected persons. In addition, babies born to untreated mothers are likely to acquire congenital syphilis during pregnancy (mother-to-child transmission, MTCT) and childbirth [2].

Syphilis diagnosis usually involves serological testing, and the disease can be easily treated with antibiotics (i.e., penicillin) since it is a bacterial infection [2]. In addition, the correct use of condoms is an effective and simple method to prevent syphilis from being spread through sexual intercourse [1]. 

Nearly 6.3 million cases of syphilis were reported among people aged 15–49 years globally in 2016 [1]. According to World Health Organization (WHO) [1], the disease disproportionately affects key populations. For instance, the incidence of syphilis is as high as 27% among men who have sex with men (MSM) and 14% among female sex workers. However, across the Americas, key populations have not been screened for syphilis in 35% of countries, even though serology tests for STIs like HIV are performed. This reinforces the fact that syphilis remains a neglected disease [1]. Additionally, WHO estimates 930,000 yearly cases of syphilis in pregnant women with active infection (transmissible during pregnancy), resulting in approximately 350,000 adverse birth outcomes annually, i.e., malformations and preterm births.

In Brazil, the 2021 Epidemiological Bulletin released yearly by the Brazilian Ministry of Health (MoH) reported that the detection rate of acquired syphilis had risen considerably until 2018, when it peaked at 76.4 cases per 100,000 population [3]. In contrast, the acquired syphilis rate dropped to 54.5 cases per 100,000 population in 2020. Of note, the infection has been listed as a notifiable disease since 2010. In 2018, the rate of congenital syphilis rate reached 9.0 cases per 1000 live births, with a drop in the following years, reaching 7.7 cases per 1000 live births in 2020 and with 186 children under one-year-old dying from syphilis, which corresponds to a mortality coefficient of 6.5 deaths per 100,000 live births in 2020. Meanwhile, the detection rate of syphilis in pregnant women reached 21.8 cases per 1000 live births in 2019 and decreased to 21.6 per 1000 live births in 2020.

In this context, public awareness campaigns play a pivotal role in disseminating relevant information on health promotion and disease prevention, especially to key populations [4,5]. Campaigns centered on themes around syphilis (i.e., transmission, prevention, symptomatology, disease progression, diagnosis, and treatment) may maximize the chance the disease is caught at early stages, thus allowing prompt and simplified treatment, and minimizing a person’s risk of becoming infected and spreading the infection.

As part of the fight against the rising incidence of syphilis in Brazil, the MoH launched the “Syphilis No!” Project (SNP). One of the main intervention actions of this project throughout the national territory was developing a massive and universal public health communication campaign throughout the country, in addition to specific actions for priority population groups.

Through the “Syphilis No!” Project the syphilis epidemic has been tackled through two strategic lines: (1) reinforcing universal actions of the National Brazilian Health System (SUS) and (2) implementing specific ones to 100 municipalities chosen by the Ministry of Health as priorities for the response to congenital syphilis, as in 2015, they represented 68.95 of the number of congenital syphilis cases in Brazil [6].

The universal line of intervention included the acquisition and distribution of supplies for testing and treatment (crystalline and benzathine penicillin), enhancing the STI laboratories network and situation rooms for epidemiological surveillance, educommunication strategies [7], social interventions, and awareness campaigns performed to face syphilis in that period [6].

The most important communication intervention of the SNP occurred between 2018 and 2019, namely the “Syphilis No!” campaign, which emphasized the concept “Test, Treat and Cure” to alert the population about the availability of the rapid syphilis test or VDRL test at any Primary Health Care (PHC) unit of the SUS.

During this period, the organizers produced and disseminated a large amount of material through television, radio, streaming platforms, printed media, magazines, posters, informative booklets, and stickers. Internet sites, specifically directed toward pregnant women, disseminated related content and other content strategically emerged within news coverage and on social networks, relationship apps, and digital pages of magazines. In addition, digital influencers made sponsored posts on their social networks [8].

Other studies have shown that the intervention actions of the SNP had positive effects, as the drop in the number of reported syphilis cases in the country and a change in the trend was observed after 2018 [6], when the Project started. Andrade et al. [9] demonstrate the hypothesis that the SNP has influenced the decline in hospitalizations for congenital syphilis in Brazilian municipalities as of May 2018. De Morais Pinto and colleagues [8] found that the SNP influenced the increase of testing nationwide, which consequently expanded diagnosis and treatment, which positively impacted the reduction of syphilis cases in the country.

Our study deepens the analysis of communication actions, especially the content of online news disseminated between 2018 and 2019. To do so, it made use of natural language processing applied to sentiment and informational content analysis of public health campaigns aimed at confronting syphilis in Brazil. The results presented confirm that the most significant increase in testing is directly related to content whose sentiment is inductive (TP3; stands for news type 3). In addition to statistically demonstrating this correlation, our results show Granger-causality between inducing sentiment content and increased testing.

Therefore, computational methods based on natural language processing that can contribute to content analysis present themselves as essential tools for studying the effectiveness of public health communication campaigns. Furthermore, such a mechanism can be applied to qualify future campaigns based on the experiences previously analyzed by it and thus better direct public efforts and investments.

This study analyzes news items disseminated online throughout the pre-, during-, and post-intervention periods of the SNP, which developed a public health campaign in Brazil between November 2018 and March 2019. In addition, we intended to understand the influence of the types of sentiment found in online news content on syphilis. Hence, we sought to identify which content encouraged people to seek testing and, by doing so, to draw possible relationships between campaign data and types of informative content disseminated along with epidemiological data.

The Agenda-setting theory [10] proposes that the readership of printed or online newspapers tends to confer more importance on subjects that obtain more prominence in journalistic coverage. The agenda-setting process operates on two levels. The first level, denominated salience, is when a topic stands out against others due to a greater volume of citations in the media in a given time interval. Hence, one can say that this topic has gained space and relevance in the media, which leads to the hypothesis that it can also gain space in the agenda of public opinion debates. If, over a period, an issue stands out in the media, it is necessary to identify how this subject is approached in journalistic coverage, that is, what the attributes are. As a result, the second level of the agenda-setting process occurs. One way this can be characterized is by the approach, whose information in the news can value negative and positive aspects about the topic or present a neutral treatment, which would only inform without highlighting negative or positive aspects.

Thus, from an Agenda-Setting theoretical perspective, the attributions of positive, negative, or neutral sentiment toward themes covered by the media are the second level of the agenda (attributes to subjects or themes/subjects). Sentiments are investigated from topics that have gained prominence/salience (first level of the agenda) in media coverage relative to other themes over a given timeframe [10].

Sentiment analysis [11] can be defined as the task of extracting subjective information about sentiments (positive, negative, or neutral) from different sources. Texts, biometric data, comments on social networks, product feedback, and others are examples of sources. This analysis allows us to know factors that influence certain social phenomena and can be used, for example, to verify the acceptance of a given product or even to understand how the target audience perceives marketing messages.

Based on a considerable body of literature, the supervised approach—that is, when there are labels assigned (in this case, a sentiment) for each element in the database—has been deemed the most applied strategy that has delivered consistent results [12,13,14]. However, building and labeling such databases is not a simple task [15]. Usually, it demands a lot of time and knowledge of the themes reported in the texts that compose the database. Thus, unsupervised strategies appear more appropriate when obtaining the database becomes a constraint.

### 1.1. Related Work

In recent years, a vast body of research has been intensively exploring sentiment analysis applied to online news. Below, we highlighted some studies found in the literature.

Lei and colleagues [16] proposed a system for sentiment detection in online news items. This system is based on document selection, part-of-speech (POS) annotations, and sentiment lexicon generation. The presented approach showed better results than a Support Vector Machine (SVM) based classifier.

Li et al. [17] introduced a Weighted Multi-label Classification Model (WMCM) for multi-label classification that applied weightings to the files used in the training process. In addition, another method was also used for checking different sentiments associated with an exact word at a semantic level.

In Bai’s work [18], sentiment analysis regarding online news was used to infer customer sentiment. The author proposes a hybrid heuristic using the Tabu search metaheuristic combined with the probabilistic Markov Envelope model to extract the dependencies between words and assemble a sentiment-based dictionary.

Rao et al. [19] proposed algorithms to automatically construct an emotion-based dictionary and a method to generate a topic-based dictionary. The method developed has some pertinent features, as it is language-independent and allows granulation, scale, and data volume adjustment.

### 1.2. Study Design

This study applied an unsupervised dictionary-based method called Valence Aware Dictionary for Sentiment Reasoning (VADER) [20], considering the Brazilian Portuguese language. Recent studies have highlighted the importance of VADER in sentiment analysis of financial news [21,22] and public health [23]. Thus, we sought to discern which sentiments were conveyed in the news we analyzed, the predominance of sentiments (polarity), and how they related to syphilis testing.

The results revealed three types of news (TP is the stand for the news type) content, called TP1 (neutral sentiment), TP2 (sentiment of security), and TP3 (attitude-inducing sentiment). Moreover, the latter causes a feeling of “vulnerability” and the public’s need to seek guidance or take action [10]. Finally, we identified that TP3 was the category that promoted syphilis testing and, therefore, treatment the most, which likely influenced the reduction of acquired syphilis, syphilis in pregnant women, and congenital syphilis.

## 2. Materials and Methods

Our experiment explored underlying sentiments in texts published as news. The idea was to find patterns and then verify whether there was a statistical correlation among them (stratified by sentiments), as well as the number of syphilis tests performed during the period the news items were published. We used a proprietary digital health information ecosystem called Hermes to develop this research. Hermes is based on a multidimensional framework for temporal analysis of public health interventions [6,8].

The VADER algorithm method was used for sentiment analysis [20], specifically an adaptation of this algorithm available at https://github.com/rafjaa/LeIA (accessed on 25 June 2021). The adaptation process involved translating the terms the algorithm used internally to Brazilian Portuguese. Our study was conducted in six stages (Figure 1): (i) Collection of the news database; (ii) Pre-processing of collected data (excluding pages with errors such as HTTP 404, 505, and news under 10 words); (iii) Application of the VADER algorithm to identify the sentiment of each news item; (iv) Categorization of the news into two groups (positive and negative), in which the neutral classification was not employed because all the news collected had the “neutral” sentiment to a greater extent, and therefore any distinction between the underlying features could not be made; (v) Correlation analysis between the group of news and the syphilis testing data; (vi) Performing the Granger test on the news and testing datasets.

### 2.1. Use of Data Sources

The database used in this study was collected by Hermes [6,8]. The ecosystem collected 1048 online pages that were Google indexed and included the term “syphilis” in Brazilian Portuguese webpages between January 2015 and December 2019, i.e., before, during, and after the “Syphilis No!” campaign. After compilation, Hermes extracted the main content of these pages, and three experts in the communication area analyzed the content in order to eliminate pages that were not news, such as search results on portals, non-textual content (documents, videos, and podcasts), and scientific content pages. From this preliminary analysis, 627 pages were classified as news. This process is fundamental because it removes irrelevant or noisy data and obtains more accurate results.

Syphilis testing data were obtained from SIA/SUS (Outpatient Information System of the Unified Health System), available on the MoH webpage (https://datasus.saude.gov.br/acesso-a-informacao/producao-ambulatorial-sia-sus/) accessed on 25 November 2020. SIA/SUS enables the download of results tests considering the amount per month and year as follows: (i) Treponemal test for syphilis detection, (ii) Fluorescent Treponemal Antibody-Absorption (FTA-ABS) IgG test for syphilis diagnosis, (iii) Fluorescent Treponemal Antibody-Absorption (FTA-ABS) IgM test for syphilis diagnosis, (iv) Nontreponemal test for syphilis detection, (v) Nontreponemal test for detecting syphilis in pregnant women, (vi) Rapid Syphilis Test, (vii) Rapid syphilis test for detecting the infection in pregnant women or fathers/partners. The testing data comprise the years 2015 to 2019 stratified by month.

### 2.2. VADER Algorithm

The VADER algorithm is designed to measure both polarity and intensity of sentiment in texts. To do this, such a model uses a set of grammar rules, syntactic precepts, and a lexeme dictionary. According to Hutto, C. & Gilbert, E. [20], during the validation process, when a set of social media text snippets (tweets) was used, the VADER algorithm obtained a better classification accuracy rate (F1 = 0.96) compared to individual human raters (F1 = 0.84), at correctly classifying the sentiment of the tweets into positive, neutral or negative classes.

The dictionary used by VADER can include words, phrases, and emoticons. Each lexeme in the dictionary is accompanied by a score that indicates the polarity and intensity of the sentiment. Hence, a negative sentiment is scored between −4 and −1. Then, 0 is assigned to a neutral word, while a positive word is scored from 1 to 4. Scores were obtained by taking the average score of 10 independent raters.

The rules used to modify punctuations when the algorithm is applied to new texts comprised: (i) Punctuation—the exclamation mark increases the intensity of the sentiment; (ii) Capital letters—words written in capital letters also increase the intensity of the sentiment; (iii) Modifiers of degree—these are words that increase or decrease the intensity of a given sentiment, e.g., “extremely”; (iv) Contrastive conjunction—changes the polarity of the sentence, making the text that follows the conjunction the dominant sentiment; and (v) Trigrams—are used to identify negations that invert the polarity of the text.

Once the algorithm evaluates the text, the score of each part that indicates a sentiment is summed and then normalized between −1 and 1 to show the overall sentiment of the text. Words not in the lexical dictionary are evaluated as having neutral sentiments (score 0). The function used to normalize the sum of the scores is shown as follows.
$$\frac{x}{\sqrt{x^{2}+\alpha }}$$
where *x* represents the sum of sentiments and *α* = 15 in the experiments performed. VADER also indicates the percentage of positive, negative, and neutral sentiments in the input text.

As an observation of the method’s limitation, both the sentiment classification [20] and the proposition of the second-level attribute of agenda-setting [10,24] are relevant for the development of the analysis. However, the attribute or sentiment being named as “negative” does not actually depict what the excerpts analyzed represent as a journalistic construction since they are informative texts. What the binary classification recognizes as data that characterize “negative” sentiment/attribute are, in fact, sentences that do not feature sensationalist, stereotypical, or stigmatizing language but information about the severity of syphilis and its consequences, implications, and concrete risks in the lives of people affected by the infection.

In order to better understand the core of the content processed, three specialists, doctors in public health communication with more than ten years of experience in this area, conducted an exploratory analysis based on the results of processing the news carried out by the VADER algorithm. These specialists identified the main characteristics of the news classified as positive and negative. The result of this analysis is summarized in Table 1.

Given this fact, for the purposes of this article, methodologically, the classifications recognized by the literature as neutral, positive, or negative sentiment were categorized by the type of the content’s meaning: TP1 (neutral sentiment), TP2 (sentiment of security), and TP3 (attitude-inducing sentiment) respectively (Table 1).

### 2.3. Statistical Methods

Some correlation methods require an evaluation of the distribution of the data. Thus, the Shapiro-Wilk test [25] was used on the news and test sentiment data. The Shapiro-Wilk test has as its null hypothesis that the distribution of the data comes from a normal. With this test, it is possible to say that a data set has a normal distribution if the *p*-value resulting from the test is greater than or equal to 0.05.

To assess the existence of a correlation between the news and test data, Spearman’s Rank Correlation Coefficient (Spearman’s rho) was used. Its values are assigned based on a range from −1 to +1, where negative values indicate a contrary correlation, i.e., when one variable grows, the other decreases. Positive values indicate a directly proportional correlation; that is, when one variable grows, the other follows suit. Finally, an index equal to zero indicates that the data sets are not monotonically correlated.

In order to reinforce the association results found using Spearman’s rho, it was also proposed to use the Granger Causality test [26]. To perform this test, it was first necessary to check if the series is stationary; for this, the Kwiatkowski–Phillips–Schmidt–Shin (KPSS) Test was used [27], using as a null hypothesis that data are stationary around a deterministic trend.

Once the series is stationary, one can proceed with the Granger Causality test [26]. This test has the principle of indicating whether one series is temporally related to another, that is, given that series X is a better predictor of Y than Y’s past. The null hypothesis of the test is that a series X does not Granger-cause series Y, so if the *p*-value is less than 0.05, the null hypothesis is rejected.

Thus, the correlation between each type of test and the amount of news types TP1, TP2, and TP3 was verified between (i) 2015 to 2017 and (ii) 2018 to 2019.

## 3. Results

The VADER algorithm was applied to the 627 news items analyzed. We found that neutral sentiments (TP1) were predominant in each of them. This result was already expected since the news in the base is informative text.

From this finding, the news was then analyzed between the TP2 (security sentiment) and TP3 (attitude-inducing sentiment) polarities. Of the 627 news items, 198 had a percentage of sentiments that led to a sentiment of security (TP2) to a greater degree, while 429 featured sentiments that instilled vulnerability (TP3) in a greater percentage when compared exclusively to the percentage of sentiments of security. In sum, approximately 31.58% of the news had TP2 features, while 68.42% had TP3.

A second analysis was performed on syphilis testing data using the reports from Outpatient Information System. Thus, the data comprised 2015 to 2019, stratified by month. It was possible to observe that between the years 2015 to 2017, the number of news items did not exceed 20 per month (Figure 2), whereas, in 2018 and 2019, considerable growth in the volume of monthly news items on the subject was observed (Figure 3).

We applied Spearman’s rho to investigate the statistical evidence of a correlation between the quantity of news (grouped by TP2 and TP3) and the number of syphilis tests performed monthly (Figure 4). Before evaluation through Spearman’s rho, the normality test was applied to the data of news and syphilis testing.

For the news data set, the normality test showed a *p*-value = 2.5 × 10^−6^, in which case the null hypothesis is rejected. For the syphilis testing dataset, the Shapiro-Wilks test resulted in a *p*-value = 0.46, so the null hypothesis cannot be discarded, meaning that the data have a normal distribution. In addition to this test, a QQ-Plot plot was generated to visualize the data distribution. Figure 4 reveals findings consistent with the statistical tests since, for the syphilis testing data, the points are distributed along the 45-degree straight line, indicating its approximately normal nature. Meanwhile, the news data shows a significant deviation from the red line, evidencing that the underlying distribution is not normal.

Knowing that the data do not have a normal distribution, the Spearman’s rho index [28] was adopted since (i) this indicator does not require the normality of both data sets, (ii) there is a monotonic relationship between the data, (iii) both variables are ordinal at least, and (iv) the observations are paired. Figure 5 depicts the scatter plot of news and syphilis testing data, thus highlighting the monotonic relationship of the variables.

The KPSS test revealed that the null hypothesis could not be rejected for both series (news items and syphilis testing), with *p*-value = 0.1 for both series at the significance level of 0.05, indicating that the series is stationary.

The calculation of the correlation test generated the following conclusions: for the TP2 type news, in all periods, the *p*-values were greater than 0.05, thus generating inconclusive results; regarding the TP3 type news in the years 2015 to 2017, there was a correlation of ρ = 0.33 between TP3 type news items and syphilis testing data (*p*-value = 0.04); the years 2018 and 2019 showed a correlation ρ = 0.67 between TP3 news and the number of syphilis tests performed per month, with *p*-value = 0.0003.

We performed a second test to ratify the robustness of the conclusions previously found. Therefore, the Granger Causality test was applied between the TP3 news data and the testing data, considering 2018 and 2019. The test was performed with lags between 1 and 5 and presented statistically significant results with a lag of 5 using the Chi-squared test and Likelihood ratio *p*-values of 0.000 and 0.002, respectively, reinforcing the association results found by Spearman’s rho.

The correlation between testing and news was also evaluated, subdividing the set by the type of syphilis screening test. It was concluded that the number of FTA-ABS IgM tests performed suggested a meaningful correlation of ρ = 0.73 (*p*-value = 3.8 × 10^−5^) and ρ = 0.77 (*p*-value = 9.6 × 10^−6^) with the number of TP2 and TP3 news items, respectively. The other test types had weaker or inconclusive correlations.

## 4. Discussion

The processes of news production on events filter the facts occurring each day according to what is called “news values” or “newsworthiness” [29]. The journalistic routines establish an order of relevance that filters from the totality of daily facts that, through journalistic screening, gain space on the pages of newspapers, TV screens, radio broadcasts, and the pages of journalistic websites on the Internet. In the decision-making process, there is a feeling that the facts selected by the media agenda were selected because they are relevant to society. To promote public health policies, public agents also play a pivotal role by providing information to those who have the power to determine the journalistic relevance [24] of a topic, e.g., the fight against syphilis.

One of the best-known ways is the investment in communication advertising campaigns and actions in the form of press consultancy, either by sponsoring news (editorials) or by inducing spontaneous coverage from the subsidy of information considered socially significant. The results of the research point out that in the comparison between 2015 to 2017 (before the “Syphilis No!” Project) and the period 2018–2019, in which the advertising campaign developed and conducted by the Project was carried out, there was a considerable increase in the monthly and annual volume of news reported in the press, which was captured by the survey. Therefore, there is a corresponding increase in the agenda-setting of syphilis, which gains significant “salience” (first-level agenda-setting proposed by McCombs) in 2018–2019 compared to 2015–2017. The increase was of the order of 520%. Thus, it is possible to infer from the results that public agents succeeded in strategically subsidizing the media with information to emphasize the news value of facts related to the problem of syphilis in Brazil.

Media communication can be an essential ally in inducing public health policies if we understand that “the greater the need for guidance that people have in public affairs, the more likely they are to pay attention to the media agenda” [24]. From the perspective of the agenda-setting theory, besides the topic of “syphilis” having gained greater salience on the media agenda, the success of the action can also be perceived when this relevance (salience) given by the media to a theme, as in the case of syphilis, finds correlation with the sense of significance and search for guidance in the perception of the individual [24]. “In general, it can be stated that a greater need for guidance around an issue causes a greater ‘vulnerability’ or receptivity to the effects of the media agenda” [10]. The results of the study at hand demonstrate that it is possible to identify the second level of agenda-setting in the analysis presented on the “attributes” [10,24] from the study of the content types and neutral (TP1), security (TP2) and attitude-inducing (TP3) sentiments, identified in the texts of the news published in both periods (2015–2017 and 2018–2019).

The results of Spearman’s rho point out that in 2018 and 2019, there was ρ = 0.67 correlation between TP3 news and the number of syphilis tests performed per month. Interestingly, the results (when analyzing the content of the news stories that were classified with TP2 or TP3 sentiment) demonstrate that both the sentiment classification [20] and the second-level attribute proposition of agenda-setting [10,24] are pertinent but are limited by only being able to classify the news stories with attribute or sentiment in a binary logic of “negative” versus “positive”. In line with studies of communication and journalism in health, with this research, it was possible to deduce that what news items present as feature types are neutral (TP1), security (TP2), and attitude-inducing sentiments (TP3), contributing with a new and productive interpretation to the attributes proposed by McCombs [24] and Ardélvol-Abreu et al. [10]. In this perspective, the news identified with attitude-inducing sentiment is precisely those constructed with grounded information about the seriousness of syphilis and its consequences, implications, and concrete risks in the lives of people affected by the infection.

It is not about returning to an outdated approach of exploring syphilis and other STIs through the discursive logic of fear and stigmatization [30] that seems to be implicit in the limited understanding of the binary sense of “negative versus positive” sentiment. Fundamental is to understand that there is a correlation between news that produces an attitude-inducing sentiment and therefore increased syphilis testing. That is because they are journalistically qualified and present the severity, consequences, implications, and risks of syphilis for the general public, especially for pregnant women and their babies, resulting in content that can generate a sense of “vulnerability” and the need for people to take action [10,24]. Thus, it encourages them to seek professional guidance and syphilis testing in health services.

From the results obtained in this study, we highlight the need to look at the hidden content of online news and extract the predominant sentiments aiming to find patterns and correlations, instigating public health policymakers to enhance the communication message in order to induce changes in the population’s behavior in response to public health crises, as in the case of the syphilis epidemic in Brazil, which favored an increase in testing to qualify the diagnosis, treatment, and cure.

Furthermore, it is noteworthy that these scientific findings can contribute to the definition of other communication strategies in public health, increasing effectiveness in coping with health crises, such as the syphilis epidemic in Brazil.

As for the limitations of this study, we can mention that even with the statistical tests that suggest the correlation between the TP3 and the search for testing, we understand that this analysis alone is not enough to determine that the search for testing was only due to the agenda-setting of the theme. Other survey methods, such as a questionnaire applied to health service users to identify why people seek syphilis testing, could be used in future research. In addition, the database used did not contain data on comprehensiveness and to which audience the news was directed, making it impossible to assess which type of populations were subsequently impacted. Data on the profile of the test population (age, sex, socioeconomic variables, ethnicity, etc.) were also not available. However, these limitations do not compromise the study results because it is important to emphasize that it is rare to have reliable data that helps measure the impact of public health actions on health-related perceptions and behaviors. Therefore, the work presented, along with its possible derivations and lines of continuity, constitutes a precious element when it comes to justifying to the responsible bodies or agencies the carrying out of public health campaigns.

## 5. Conclusions

This multidisciplinary study analyzed online news published during the syphilis epidemic in Brazil, highlighting the importance of using computational methods to extract predominant sentiments in the textual content and identifying what type of news induces the increase in testing.

## Figures and Tables

**Figure 1 ijerph-19-16801-f001:**
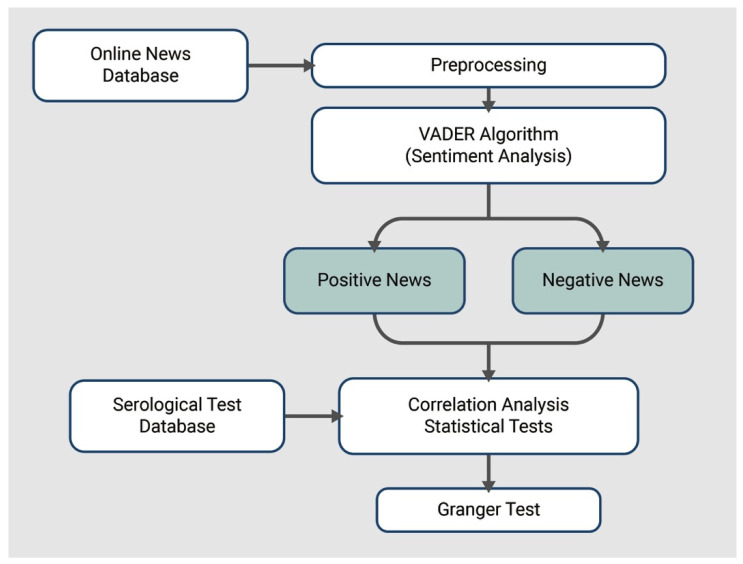
Sequence of stages carried to the classification of each news in their sentiment polarity, as well as the correlation evaluation and Granger Causality Test.

**Figure 2 ijerph-19-16801-f002:**
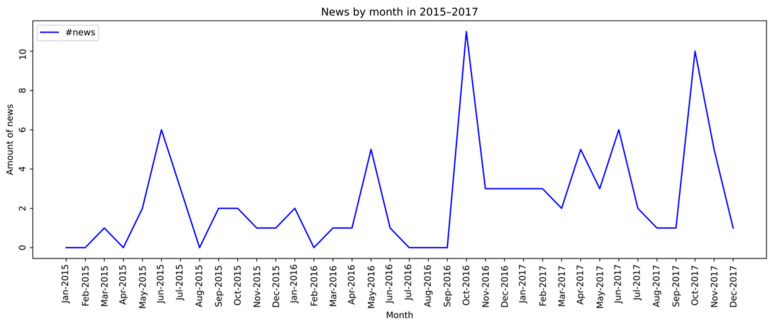
News month by month between 2015 and 2017 about syphilis.

**Figure 3 ijerph-19-16801-f003:**
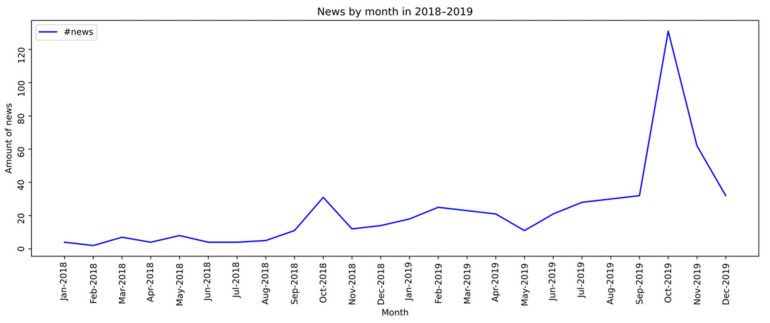
News month by month between 2018 and 2019 about syphilis.

**Figure 4 ijerph-19-16801-f004:**
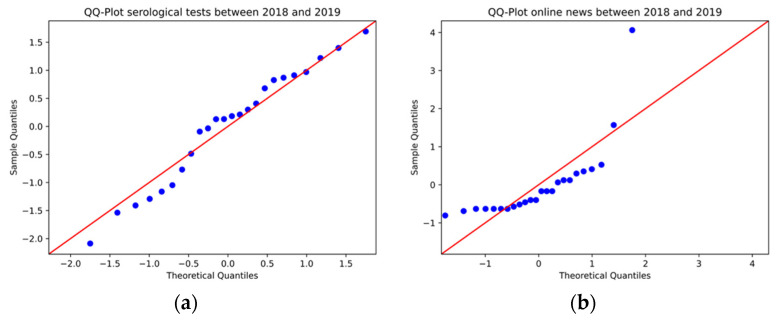
QQ-Plot of (**a**) syphilis testing and (**b**) online news items between 2018 and 2019.

**Figure 5 ijerph-19-16801-f005:**
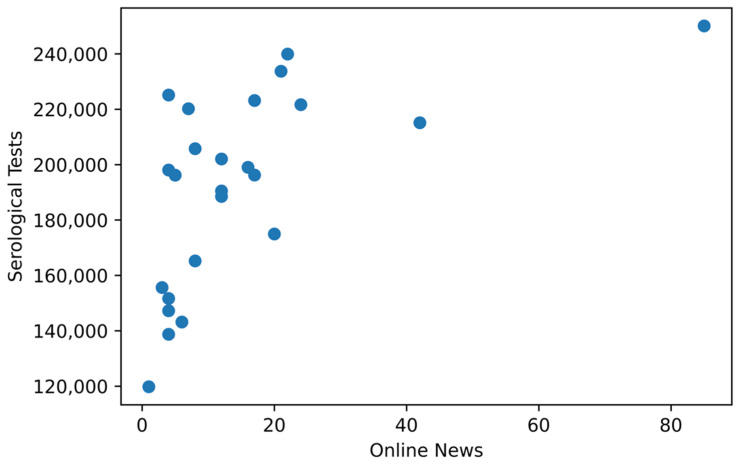
Scatterplot of news and testing data showing the monotonic shape of the relationship between news and testing data.

**Table 1 ijerph-19-16801-t001:** Terminology proposed based on the content analyzed in the news, according to characteristics proposed by Ardèvol-Abreu et al. [10].

Label	Polarity	Sentiment	Context in This Study
TP1	Neutral	Neutral	Descriptive and informative information with relevant data but without relevant emotional appeal
TP2	Positive	Security	Descriptive and informative information from relevant data, but with content that produces a sense of security, thus not instilling a need for guidance in the audience
TP3	Negative	Attitude-inducing	Substantiated information about the seriousness of syphilis and its consequences, implications, and actual risks in the lives of people affected by the infection, thus generating in the audience a feeling of “vulnerability” and the need to seek guidance or take some action

## Data Availability

The data sets used and analyzed during the current study are available from the corresponding author on reasonable request.

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
