# Peer review of "Sentiment Analysis in Understanding the Potential of Online News in the Public Health Crisis Response"

_ijerph, 2022, doi:10.3390/ijerph192416801_

Round 1
Reviewer 1 Report
Within this work, the authors analyze online news disseminated throughout the pre-, during-, and post-intervention periods of the "Syphilis No!" Project. This analysis allowed them to identify salience and attributes (neutral TP1, security TP2, and attitude-inducing sentiments TP3). Then, the authors analyze the correlations between these news and the number of syphilis tests performed.
The final aim of the authors is to analyze the role of media and communication (in particular those of the T3 category) in determining healthcare prevention behaviors in citizens.
Overall, I believe there is a strong structural weakness of the article.
Indeed, the authors identify a correlation between TP3 news and the number of syphilis tests performed per month. The number of tests and TP3 news are correlated both in the period 2015-2017 (pre "Syphilis No!" Project) and in 2018-2019 (during "Syphilis No!" Project). However, the fact that there is a correlation between news and tests carried out is quite intuitive. However, this is not enough to provide sufficient added value.
Furthermore, it should be considered that if on the one hand it is true that news influence the quantity of tests, the opposite could also happen. In fact, the very same case reported by the authors (see positive case in Panel 1) is the perfect example: the news reported is a consequence of the screening action by the municipality of Santos, therefore the increase in tests has resulted in the generation of new online content.
From this point of view, the authors try to infer a causal relationship through the Granger Causality test. However, this test was only applied in the period 2018-2019, by which time the project was already in place. Regrettably, if both variables are influenced by a third variable (namely the presence of the "Syphilis No!" Project), the Granger test loses its meaning. It would have been useful to know if this causality is also present in the 2015-2017 period, but this type of analysis has not been done.
In general, I fear that the authors' hypotheses do not make a sufficient contribution to the state of the art and are not sufficiently supported by the data.
Reviewer 2 Report
Thank you for a very comprehensive and interesting article, which is timely not only for the work being done in Brazil to fight STIs like Syphilis, but for the global community, that deals with the same STIs.
The introduction is strong overall: but more clarification or maybe a definition could be used to tell the reader exactly what TP terminology stands for see line 99? ( terminology proposed???) you use an abbreviation that is never spelled out in the text. The authors do a great job of explaining what each TP means but not what the T and P stand for. That would help early on.
This article is extremely technical in its material and methods, statistical analysis, and a strong discussion. That is a good thing for someone that may want to use the article and study to recreate or test the significance of a public health campaign so thank you for that.
line 464-471 the IRB section needs some clearing up and important information added that was not in the body of the text nor in the IRB section. That is key to your study.
Round 2
Reviewer 1 Report
I'm afraid the authors' response confirms my doubts. Both news and tests are influenced by a third variable (namely the presence of the "Syphilis No!" Project) which is however not taken into account in the study.
I believe that the proposed results are not supported by the authors' analysis and that they do not make a sufficient contribution to the state of the art.
Author Response
Please see the attachment.
Reviewer 1
2 - Comments and Suggestions for Authors
I'm afraid the authors' response confirms my doubts. Both news and tests are influenced by a third variable (namely the presence of the "Syphilis No!" Project) which is however not taken into account in the study.
I believe that the proposed results are not supported by the authors' analysis and that they do not make a sufficient contribution to the state of the art.
Response:
Dear reviewer, we appreciate your comments and recognize that your question is pertinent and perhaps some essential aspects of the "Syphilis No!" Project were not clear in the text.
Through the "Syphilis No!" Project, the syphilis epidemic has been tackled through two strategic lines: (1) reinforcing universal actions of SUS and (2) implementing specific ones to 100 municipalities chosen by the Ministry of Health as priorities for the response to congenital syphilis, as in 2015 they represented 68.95% of the number of congenital syphilis cases in Brazil [6].
The universal line of intervention included the acquisition and distribution of supplies for testing and treatment (crystalline and benzathine penicillin), enhancing the STI laboratories network and situation rooms for epidemiological surveillance, educommunication strategies [7], social interventions, and awareness campaigns performed to face syphilis in that period [6].
The "Syphilis No!" campaign (carried out from November 2018 to May 2019) conveyed the clear message of "Test, Treat and Cure", to alert the population about the availability of the rapid syphilis test or VDRL test at any Primary Health Care (PHC) unit of the National Brazilian Health System (SUS) [8].
During this period, the organizers produced and disseminated a large amount of material through television, radio, streaming platforms, printed media, magazines, live broadcasts during events, posters, informative booklets, and stickers. Internet sites specifically directed toward pregnant women disseminated related content, and other content strategically emerged within news coverage and on social networks, relationship apps, and digital pages of magazines. In addition, digital influencers made sponsored posts on their social networks.
This information in red has been added to the introduction, along with its references.
Thus, the main contribution of this work is to analyze and compare the prevailing sentiment of online news about syphilis (within the context of an epidemic in Brazil) in pre- and post-campaign periods in order to direct the efforts of decision-makers better when designing public health policies and new public health intervention projects, as well as effectively intervening and changing the population's behavior aiming to avoid the increase of the public health crisis. In this case, the syphilis epidemic in Brazil.
Thus, it is not about adding value to the state of the art of computational methods based on natural language processing in computer science, such as sentiment analysis. Instead, this study adds value by applying this technique, analyzing its results, and demonstrating the correlation between screening tests performed by SUS and online news with common characteristics (TP3) that induce the population to seek treatment.
Therefore, the significant contribution lies in applying the method and how it can contribute to conducting public health policies in the specific area of ​​massive public health communication campaigns. This is a relevant scientific finding for public health, as it has been shown that specific contents can better induce or promote the engagement of the population, which in this case of the increased testing, an aspect that favors the treatment and cure of syphilis, acting to mitigate the transmission curve effects.
From the results obtained in this study, we highlight the need to look at hidden contents of online news and extract the predominant sentiments aiming to find patterns and correlations, instigating public health policymakers to enhance the communication message in order to induce changes in the population's behavior in response to public health crises, as in the case of the syphilis epidemic in Brazil, which favored an increase in testing to qualify the diagnosis, treatment, and cure.
Furthermore, it is noteworthy that these scientific findings can contribute to the definition of other communication strategies in public health, increasing effectiveness in coping with health crises, such as the syphilis epidemic in Brazil.
These last two paragraphs (in red) were also added to the end of the discussion section of the manuscript.
